# Evaluation of the diagnostic performance of laboratory-based c-reactive protein as a triage test for active pulmonary tuberculosis

Thomas H. A. Samuels[1,2☯¤a], Romain Wyss[3☯], Stefano Ongarello[3], David A. J. Moore[1,2], Samuel G. Schumacher[3‡]*, Claudia M. Denkinger[3¤b‡]

**1** London School of Hygiene and Tropical Medicine, London, United Kingdom, **2** Hospital for Tropical Diseases, University College London Hospitals NHS Foundation Trust, London, United Kingdom, **3** Foundation for Innovative New Diagnostics (FIND), Geneva, Switzerland

☯ These authors contributed equally to this work.
¤a Current address: University College London Hospitals NHS Foundation Trust, London, United Kingdom
¤b Current address: Division of Tropical Medicine, University of Heidelberg, Heidelberg, Germany
‡ SGS and CMD also contributed equally to this work.
* samuel.schumacher@finddx.org

**Data Availability Statement:** All relevant data are within the manuscript and its Supporting Information Files.

## Abstract

### Introduction

A highly sensitive triage test that captures most symptomatic patients at increased likelihood of having pulmonary tuberculosis (PTB) would 'rule-out' lower-risk patients from expensive confirmatory testing. Although studies have assessed the diagnostic accuracy of a C-reactive protein (CRP) triage test for PTB in HIV+ patients, limited data are available from HIV-cohorts.

### Materials and methods

In this retrospective case-control study, 765 serum samples were selected from FIND's biobank. Each sample had been collected from an adult presenting with respiratory symptomatology to district hospitals in South Africa and referral hospitals in Cambodia, Peru, Georgia and Vietnam between 2007–2017. Serum CRP measurements were obtained using a laboratory-based assay. CRP cutoff-points of ≥8mg/L and ≥10mg/L were predefined as a positive triage test result. The PTB reference standard was two contemporaneously collected sputum liquid culture results.

### Results

CRP demonstrated an overall sensitivity for PTB of 79.8% (95%CI 75.5–83.5) and 77.7% (95%CI 73.4–81.6) for cutoff-points of 8mg/L and 10mg/L respectively. Specificity was 62.8% (95%CI 57.8–67.6%) and 66.6% (95%CI 61.1–70.7) respectively. Area-under-the-curve using Receiver Operating Characteristic analysis was 0.77 (95%CI 0.74–0.81). Threshold analysis showed optimal CRP cutoff-points were higher in HIV+ than HIV- participants. An algorithm in which CRP triage was followed by confirmatory Xpert MTB/Rif testing

**Funding:** This work was funded by the UK Department for International Development (DFID) grant number 300341-102 (https://www.gov.uk/government/organisations/department-for-international-development), Dutch Ministry of Foreign Affairs grant number PDP15CH14 (https://www.government.nl/ministries/ministry-of-foreign-affairs), Australian Department of Foreign Affairs and Trade (DFAT) grant number 70957 (https://www.dfat.gov.au/). CD received all awarded funding. The funders had no role in study design, data collection and interpretation, or the decision to submit the work for publication.

**Competing interests:** The authors have declared that no competing interests exist.

achieved a sensitivity of 75.1% (95%CI 69.0–80.4%) whilst decreasing Xpert usage by 40.6%.

## Discussion

CRP may not meet the challenge of a catch-all TB triage test. However, it shows promise in HIV+ individuals. Further research is required in a prospective study using point-of-care platforms to further evaluate its capabilities.

## Introduction

New diagnostic approaches are urgently needed to reduce tuberculosis (TB) incidence and meet WHO End TB targets [1]. The Xpert MTB/Rif assay (hereafter referred to as 'Xpert') is more sensitive than smear microscopy, faster than traditional culture methods and has been rolled out in many settings throughout the world [2, 3]. However, placement of the instrument at primary care level in low resource settings where the majority of individuals with TB first access care is restricted by its infrastructure requirements and cost [4–6]. Efforts are being made to develop Xpert platforms suited to the available infrastructure, but the cost of testing all people undergoing evaluation for presumed-TB remains prohibitive.

Currently, two weeks of cough is widely used as a prompt to initiate diagnostic testing for pulmonary TB (PTB). However, TB prevalence amongst these individuals is low [7]. Interest has therefore grown in the usage of a two-stage diagnostic algorithm in which a highly sensitive first triage test, applied to patients with symptoms of active pulmonary TB and deployed at the peripheral reaches of the healthcare system, effectively captures most subjects at higher likelihood of having TB. This triage test should also have sufficient specificity to enable 'rule-out' from the expensive second stage of testing for those at very low likelihood, creating significant resource savings especially in medium prevalence settings. The target product profile (TPP) suggested by the World Health Organization (WHO) for such a triage test specifies a minimum sensitivity of 90% and specificity of 70% against the confirmatory test for PTB [8].

C-reactive protein (CRP) is an attractive candidate as a triage test for TB. Many point-of-care (POC) CRP tests are commercially available with some costing less than US$1 [9], and this blood-based biomarker has shown promise in the diagnosis of subjects undergoing evaluation for TB. A recent meta-analysis has shown that CRP has a high sensitivity (93%) and moderate specificity (60%) amongst outpatients with PTB symptoms in high-burden settings, with data primarily coming from studies in people living with HIV [10]. Also, it has been shown to be an effective screening test for HIV+ patients and could reduce the number of Xpert tests required without comprising diagnostic yield over symptom-based screening [11, 12]. In contrast, the evaluation of the performance of CRP as a triage test has been less well-studied. A recent meta-analysis of diagnostic accuracy studies found CRP to perform with an overall sensitivity and specificity of 89% and 57% respectively for pulmonary TB, showing significant variation by geography, HIV status and clinical setting [13]. CRP was not deployed as a triage test in any of the included studies. However, several studies in HIV+ South African patients in which CRP has been used as a triage test demonstrated similar performance [14, 15]. To our knowledge no dedicated studies have done so in HIV- patients. Individuals who present to healthcare with symptoms suggestive of PTB are likely to have a greater burden of pyogenic infections and non-TB inflammatory conditions than populations being screened, which may negatively affect specificity compared with studies evaluating CRP in this context [10].

Although some studies have assessed the diagnostic accuracy of CRP in TB, no study has yet assessed the suitability of CRP for use as a triage test for individuals undergoing evaluation for presumed TB independent of HIV status in representative high endemic countries. In this study, we aimed to assess the diagnostic accuracy of quantitative laboratory-based CRP as a triage test for the detection of active TB in bio-banked serum samples taken from individuals with presumed TB presenting to secondary care across five high-burden TB countries.

## Materials and methods

### Ethics statement

All study-related activities were approved by the Human Research Ethics Committees (HREC) of the partners in-countries. These were: University of Cape Town, South Africa; Universidad Peruana Cayetano Heredia, Peru; Pham Ngoc Thach Hospital, Vietnam; Calmette Hospital, Cambodia; National Centre for Tuberculosis and Lung Diseases, Georgia.

### Study population

For this retrospective nested case-control study, serum samples were selected from FIND's biorepository. These samples were previously collected from adults presenting at district hospitals in South Africa and referral hospitals in Cambodia, Peru, Georgia and Vietnam between 2007 and 2017. Participants were included if age was greater to or equal to 18 years, if they were able to give informed consent and had presented with symptoms suggestive of pulmonary TB. Symptoms were specified as current cough, haemoptysis, recent weight loss, night sweats and fever. To be included in the analysis all participants had to provide both a blood sample and at least 2 sputum samples at enrolment prior to initiation of anti-tuberculous therapy. Blood samples were preserved in temperature-controlled freezers at -80˚C from collection until CRP testing. Participants were excluded if they had received anti-tuberculous therapy within 60 days prior to enrolment.

The study followed a nested case-control design. Cases were individuals with TB and controls were individuals presenting with symptoms that warranted investigation and for whom sputum testing proved negative for *M.tb*. The cases and controls were chosen to reflect the geographic distribution, age and sex of participants across the entire cohort and were weighted to have at least 20% smear-negative results among culture-positive TB patients at each site and at least 25% of HIV+ patients overall. Cases and controls were matched by HIV status. All study-related activities were approved by the Human Research Ethics Committees of the partners in-countries. Written informed consent was obtained from patients, as per study protocol (protocol and consent form available upon request). Study participation did not affect standard of care. Reporting followed STARD guidelines [16].

### Index test

Serum CRP levels were measured using the Multigent CRP Vario assay (a latex immunoassay) on Abbott Architect C8000 at Quest laboratories. For testing, in brief, the stored serum was thawed and then processed and interpreted, following the manufacturer's protocol.

### Reference testing

All sputum samples were processed using standardized protocols in centralized accredited laboratories of the partner institutions. The testing flow for the samples is available in the study protocol (S1 File). Testing was performed on all available sputum specimens and included smear fluorescence microscopy with Auramine staining, MGIT liquid culture (Becton

Dickinson, Franklin Lakes, USA) and solid culture on Löwenstein-Jensen (LJ) medium. The presence of *M.tb* complex in solid and liquid culture was confirmed with MPT64 antigen detection and/or the MTBDRplus line probe assays (both Hain Lifesciences, Nehren, Germany). Xpert MTB/Rif ('Xpert', Cepheid, Sunnyvale, USA) was performed on sputum samples from 2011 onwards when it was available, following the manufacturer's protocol. The operators of the index were blinded to the results of the reference standard.

## Definitions

A CRP cutoff-point of ≥8mg/L or ≥10mg/L was predefined as a positive triage test. These values were chosen based on previous work [10, 11]. To obtain the best estimate of performance of CRP, a comparison was made against a microbiological reference standard (MRS). For this MRS, participants were assigned to two diagnostic categories using a combination of laboratory and clinical findings. Cases (TBpos) included patients with any culture positive for *M.tb*. Controls (TBneg) were patients with negative microscopy, cultures and Xpert tests (when available) for *M.tb* (and at least one non-contaminated culture result), who were not started on anti-TB treatment and were alive and improved at 8-weeks follow-up. Participants who did not fulfil these criteria were excluded from selection into the study. In order to evaluate CRP against the most likely confirmatory test available in low-and middle-income country (LMIC) settings, participants were classified based on a single sputum Xpert result (Xpert reference standard; XRS). Participants with an indeterminate Xpert result were excluded from the XRS. All classification of participants was done prior to performing the index test.

## Sample size calculation

391 cases and 374 controls were sufficient to achieve a total 95% confidence interval (95%CI) width of <10% based on the targeted performances of the index test (a sensitivity of 90% and a specificity of 70% as per the minimum target product profile specified by the WHO). 212 HIV + individuals were selected amongst these, 111 cases and 101 controls, achieving a total 95%CI width of 11% for sensitivity and 23% for specificity for HIV+ participants based on the same targeted test performance.

## Statistics

For descriptive statistics, dichotomous variables were reported as numbers with percentages whilst continuous variables either as median with an inter-quartile range (IQR) if non-parametric or mean with a standard deviation (SD) if normally-distributed. The primary outcome was the overall estimates of sensitivity and specificity of CRP in the diagnosis of TB based on the MRS. Secondarily, we compared CRP against the Xpert reference standard. Results were reported for pre-defined CRP cutoff-points as point estimates with 95%CI calculated using Wilson's method [17]. An analysis of the following sub-groups was prespecified: smear status, HIV status, number of symptoms (categorized as < = 1, 2–3 or > 3). Analyses by region and gender were post-hoc based-on data trends. Receiver Operating Curve (ROC) analysis was undertaken to explore the performance of alternative CRP thresholds. Post-hoc, optimal cutoff-points were selected using a manual data inspection approach. The total combined sensitivity and specificity deficit relative to the minimum TPP was calculated for all cutoff-points along the ROC curve. The optimal cutoff-point was defined as that with the lowest combined deficit. A log-scale linear multivariable regression model of the effect of a pre-defined selection of variables on CRP concentration was generated. Variables were selected for the multivariable model if the p-value was less than 0.1 in univariate analysis (see Supporting Information). Akaike information criterion (AIC) was used to build the best model. Log

transformation of CRP concentration was used to achieve normal distribution of residuals. A complete case analysis was used. The relationship between CRP concentration and time-to-sputum culture positivity was explored using LOWESS and Pearson's correlation coefficient. All analyses were performed using STATA 15 (StataCorp, USA). An analysis plan was predefined and is available upon request.

## Results

### Participants

Of the 765 participants included in the study, 307 (40.1%) were female and the median age was 36 (IQR 27–47) (Table 1). Xpert results were available for 527/765 (68.9%) participants. Participants were drawn from five countries, with the majority being from Peru, South Africa and Vietnam (87.6% of the total). HIV status was available for 96.7% of participants, of which 28.8% were seropositive. HIV seropositivity differed markedly between study sites, ranging from 0% (Georgia and Cambodia) to 63.7% (South Africa). CD4 counts were available for 44.6% of HIV+ participants, with a median count of 202 (IQR 76–444).

The vast majority of patients presented with cough. Night sweats, fever and haemoptysis were seen more frequently in TBpos participants, and frequently co-existed (S2 Fig). Thirty percent of participants presented with 4 or more symptoms, two-thirds of whom had TB.

### Distribution of CRP

CRP varied significantly by TB status (Fig 1): the median CRP for TBpos participants was 47.1mg/L (IQR 12.3–93.0mg/L), and 4.25mg/L (IQR 1.3–19.1mg/L) for TBneg participants with a mean difference 41.3mg/L (95%CI 33.4–49.2mg/L). Similarly, CRP results were significantly higher in smear-positive TBpos participants compared to smear-negatives (median of 55.3mg/L (IQR 20.5–106.3mg/L) vs 18.7mg/L (IQR 2.2–59.3mg/L) p<0.001). Furthermore, median CRP concentration increased with the number of symptoms participants reported at presentation and was higher in HIV positive participants than HIV negative. Higher CRP concentrations were associated with shorter time to culture positivity, a proxy of mycobacterial load (S3 Fig).

The results of the multivariable regression are shown in Table 2. As expected from the CRP distribution shown in Fig 1, positive TB status and a positive smear microscopy result, as well as an increased number of symptoms at presentation, remained strongly associated with higher CRP results. Higher CRP results were also seen in HIV+ (p<0.001).

### Receiver Operating Characteristic (ROC) curve analysis

Fig 2A shows the ROC curve for data using the MRS. Area under the curve (AUC) equalled 0.77 (95%CI 0.74–0.81). CRP did not meet TPP at any cutoff-point. The CRP cutoff-point with the closest diagnostic performance to TPP was 12mg/L (sensitivity 76.0% (95%CI 71.5–79.9%), specificity 69.5% (95%CI 64.7–74.0%)). Fig 2B shows the ROC curve using the XRS. AUC equalled 0.83 (95%CI 0.79–0.87). However, CRP still did not meet the TPP at any cutoff-point using the XRS. A cutoff of 18mg/L gave a diagnostic performance closest to the TPP (sensitivity 84.7% (95% CI 78.8–89.2%), specificity 69.5% (95%CI 64.4–74.1%)). The effect of varying the CRP cutoff-point on diagnostic performance against the MRS and XRS is shown in the S3 and S5 Tables.

### Sensitivity and specificity analysis

At a cutoff-point of 8mg/L, CRP concentrations were elevated in 451/765 participants including 312/391 participants with culture-confirmed TB. This corresponded to a sensitivity of 79.8% (95%CI 75.5–83.5). Using a cutoff-point of 10mg/L CRP concentrations were found to

**Table 1. Participant characteristics.**

| Participant Characteristics | | N (total participants = 765) | |
|---|---|---|---|
| | | TBpos (n = 391) | TBneg (n = 374) |
| Age | Median (IQR) | 33 (25–43) | 39 (31–54) |
| Sex | Female (%) | 125 (32.0) | 182 (48.7) |
| | Male (%) | 266 (68.0) | 192 (51.3) |
| Site of Study | Cambodia (%) | 30 (7.7) | 18 (4.8) |
| | Georgia (%) | 30 (7.7) | 17 (4.5) |
| | Peru (%) | 125 (32.0) | 138 (36.9) |
| | South Africa (%) | 70 (17.9) | 70 (18.7) |
| | Viet Nam (%) | 136 (34.8) | 131 (35.0) |
| Smear status | S+C+ (% of TB cases) | 289 (73.9) | |
| | S-C+ (% of TB cases) | 102 (26.1) | |
| HIV status | Positive (% of participants with known HIV status) | 111 (28.8) | 102 (28.7) |
| | Negative (% of participants with known HIV status) | 274 (71.2) | 253 (71.3) |
| | Unknown (% of total) | 6 (1.5) | 19 (5.1) |
| CD4 Count | <200 (% with CD4 count) | 29 (61.7) | 18 (37.5) |
| | > = 200 (% with CD4 count) | 18 (38.3) | 30 (62.5) |
| | Unknown (% of HIV+ participants) | 64 (57.7) | 54 (52.9) |
| | Median CD4 count (IQR) | 128 (71–357) | 279 (106–515) |
| History of BCG | Positive history of vaccination or scar present (%) | 209 (53.5) | 206 (55.1) |
| | Negative history of vaccination (%) | 78 (19.9) | 83 (22.2) |
| | Unknown history of vaccination and scar indeterminate (%) | 74 (18.9) | 68 (18.2) |
| | Not obtained (%) | 30 (7.7) | 17 (4.5) |
| Prior history of TB | Positive (%) | 75 (19.2) | 96 (25.7) |
| | Negative (%) | 312 (79.8) | 275 (73.5) |
| | Unknown (%) | 4 (1.0) | 3 (0.8) |
| QuantiFERON result | Positive (% with result) | 72 (73.5) | 0 (0) |
| | Negative (% with result) | 20 (20.4) | 120 (85.1) |
| | Indeterminate (% with result) | 6 (6.1) | 21 (14.9) |
| | Not obtained (% total participants) | 293 (74.9) | 233 (62.3) |
| Number of symptoms at presentation | 1 (%) | 41 (10) | 118 (32) |
| | 2–3 (%) | 195 (50) | 183 (49) |
| | 4+ (%) | 155 (40) | 73 (19) |
| Symptoms at presentation | Cough (%) | 387 (99) | 363 (97) |
| | Haemoptysis (%) | 88 (23) | 37 (10) |
| | Fever (%) | 237 (61) | 169 (45) |
| | Night Sweats (%) | 227 (58) | 144 (39) |
| | Recent weight loss (%) | 142 (36) | 164 (44) |
| GeneXpert MTB/Rif status (n = 527) | Xpert+ (%) | 183 (83) | 0 (0) |
| | Xpert- (%) | 38 (17) | 306 (100) |

be elevated in 431/765 participants, including 304 with culture-confirmed TB, corresponding to a sensitivity of 77.7% (95%CI 73.4–81.6). The corresponding specificity was 62.8% (95%CI 57.8–67.6) using an 8mg/L cutoff-point and 66.6% (95%CI 61.1–70.7) using 10mg/L (Table 3). Against the XRS, CRP showed an overall sensitivity of 90.7% (95%CI 85.6–94.1) and a specificity of 57.3% (95%CI 52.0–62.4) at the 8mg/L cutoff-point (S2 Table). At a 10mg/L cutoff-point this was 89.6% (95%CI 84.4–93.3) and 60.8% (95%CI 55.5–65.8) respectively.

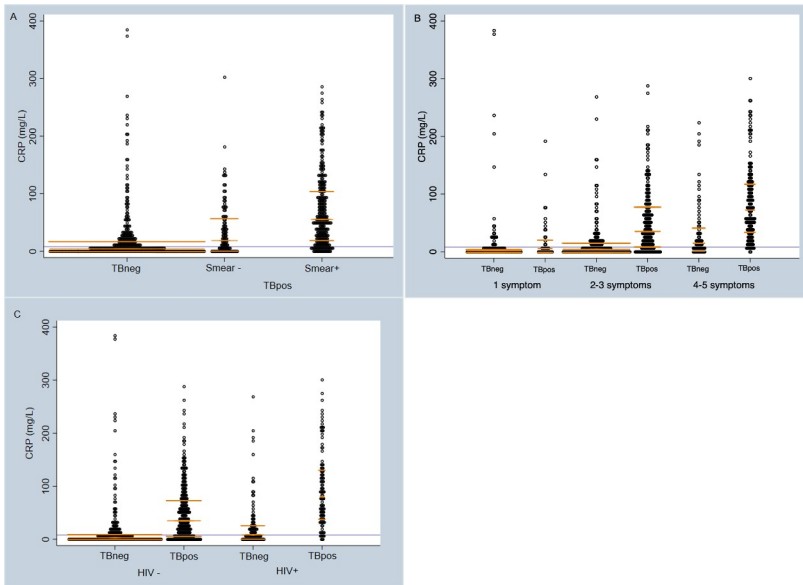

**Fig 1. The distribution of CRP amongst all study participants (n = 765).** A) by TB and smear status; B) by TB status and number of symptoms at presentation; C) by TB and HIV status. Lavender line demarcates a CRP threshold of 8mg/L–subjects below this line would not go forward for further testing in a triage-based algorithm. Gold lines demarcate median and inter-quartile range. TBneg = tuberculosis negative by MRS. TBpos = tuberculosis positive by MRS. HIV = human immunodeficiency virus. CRP = C-reactive protein.

## Subgroup analyses

Diagnostic performance against the MRS was analysed for pre-specified sub-groups (Table 3). CRP became more sensitive but less specific as the number of symptoms at presentation increased. It showed greater sensitivity for smear-positive over smear-negative TB (86.2% (95%CI 81.7–89.7%) vs. 61.8% (95%CI 52.1–70.6%) at a cutoff-point of 8mg/L). CRP was highly sensitive for TB in HIV+ participants, showing a sensitivity of 93.7% (95%CI 87.6–96.9%) and 91.9% (95%CI 85.3–95.7%) at cutoff-points of 8mg/L and 10mg/L respectively (Table 3). However, specificity was suboptimal at 49.0% (95%CI 39.5–58.6%) and 52.9% (95% CI 43.3–62.3%) respectively. By contrast, in HIV- participants CRP was less sensitive but was markedly more specific, with a sensitivity of 74.8% (95%CI 69.4–79.6%) and specificity of

**Table 2. Log-scale linear regression model of the effect of shown variables on CRP concentration.**

| Variable | | Log median fold-change in CRP (from reference) | 95% Conf. Interval | | P value |
|---|---|---|---|---|---|
| | | | Lower | Upper | |
| TB status | TBpos | 2.12 | 1.51 | 2.94 | <0.001 |
| | TBneg | Reference | | | |
| Smear microscopy result | Smear+ | 2.60 | 1.85 | 3.67 | <0.001 |
| | Smear- | Reference | | | |
| HIV status | HIV+ | 2.06 | 1.62 | 2.61 | <0.001 |
| | HIV- | Reference | | | |
| Number of symptoms at presentation | 4+ | 4.46 | 3.22 | 6.18 | <0.001 |
| | 2–3 | 2.09 | 1.57 | 2.78 | <0.001 |
| | 1 | Reference | | | |
| n = 735 | Prob > F: <0.0001 | AIC = 2661 | R-squared = 0.3817 | | |

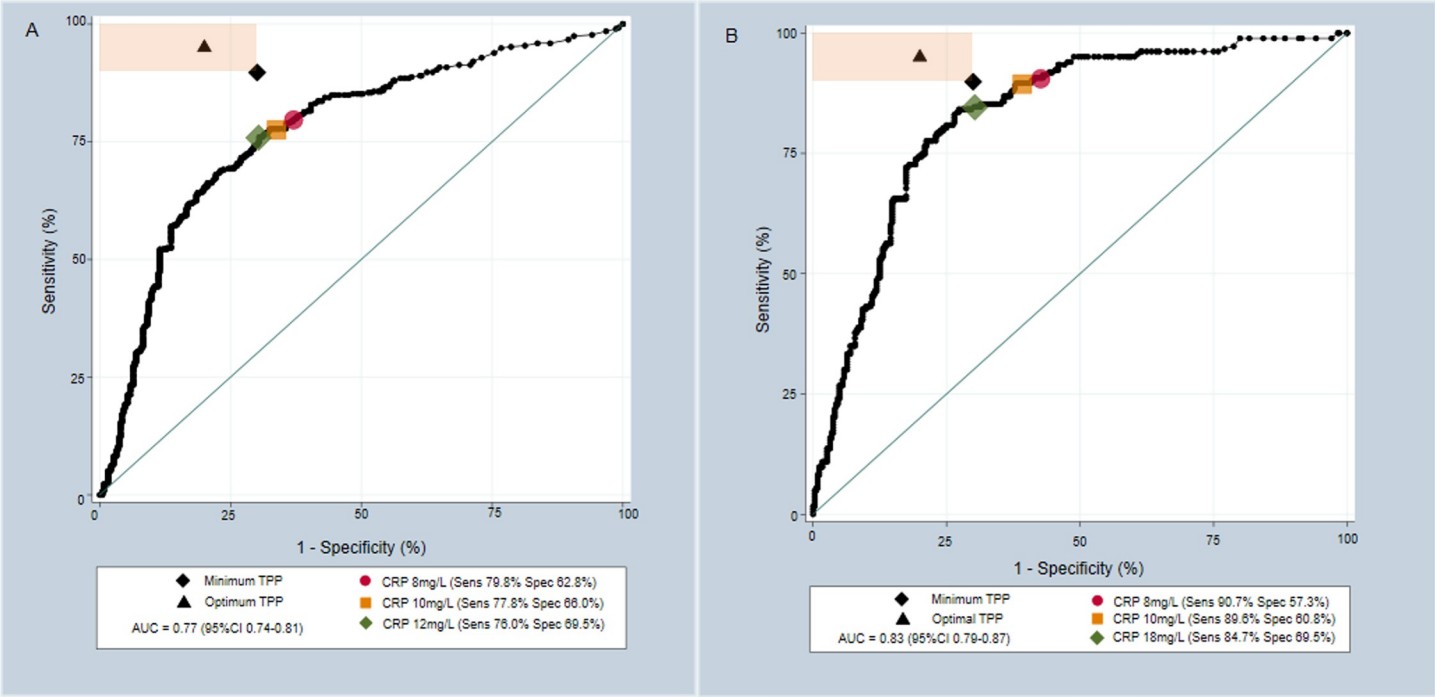

**Fig 2. Receiver Operating Characteristic (ROC) curves demonstrating the performance of CRP in the diagnosis of PTB.** A) ROC curve against culture; B) ROC curve against Xpert MTB/Rif. The shaded orange area represents the sensitivity and specificity combinations that meet at least the minimum target of one of the Target Product Profile characteristics. Minimum (Sensitivity 90% Specificity 70%) and Optimal (Sensitivity 95% Specificity 80%) Target Product Profile targets are plotted. Pre-defined CRP cutoff-points are plotted as red circles and yellow squares with optimal CRP cutoff-points plotted as green diamonds; all have estimated performance stated. AUC = Area Under Curve; TPP = Target Product Profile; PTB = Pulmonary tuberculosis.

68.0% (95%CI 62–73.4%) at a cutoff-point of 8mg/L, and 72.6% (95%CI 67.1–77.6%) and 71.1% (95%CI 65.3–76.4%) at a cutoff-point of 10mg/L respectively.

The optimal cutoff-point was markedly different in HIV+ and HIV- participants, although the difference in the AUC between these two populations was not significant. For HIV-

**Table 3. Sensitivity and specificity of CRP in the diagnosis of culture-positive tuberculosis.**

| Variable (n/N) | | CRP ≥8mg/L | | CRP ≥10mg/L | | AUC (95%CI) |
|---|---|---|---|---|---|---|
| | | Sensitivity (95%CI) | Specificity (95%CI) | Sensitivity (95%CI) | Specificity (95%CI) | |
| Overall *(391/765)* | | 79.8 (75.5–83.5) | 62.8 (57.8–67.6) | 77.7 (73.4–81.6) | 66.0 (61.1–70.7) | 0.77 (0.74–0.81) |
| Smear Status | | | | | | |
| *(289/663)* | Smear + (Culture+) | 86.2 (81.7–89.7) | 62.8 (57.8–67.6) | 84.4 (79.8–88.2) | 66.0 (61.1–70.7) | 0.82 (0.79–0.85) |
| *(102/476)* | Smear—(Culture+) | 61.8 (52.1–70.6) | 62.8 (57.8–67.6) | 58.8 (49.1–67.9) | 66.0 (61.1–70.7) | 0.63 (0.57–0.70) |
| HIV status | | | | | | |
| *(111/213)* | Positive | 93.7 (87.6–96.9) | 49.0 (39.5–58.6) | 91.9 (85.3–95.7) | 52.9 (43.3–62.3) | 0.83 (0.78–0.89) |
| *(274/527)* | Negative | 74.8 (69.4–79.6) | 68.0 (62.0–73.4) | 72.6 (67.1–77.6) | 71.1 (65.3–76.4) | 0.76 (0.72–0.81) |
| Number of presenting symptoms | | | | | | |
| *(41/159)* | 1 symptom | 43.9 (29.9–59.0) | 78.8 (70.6–85.2) | 34.1 (21.6–49.5) | 81.4 (73.4–87.4) | 0.61 (0.51–0.71) |
| *(195/378)* | 2–3 symptom | 76.4 (70.0–81.1) | 62.8 (55.6–69.5) | 75.4 (68.9–80.9) | 66.1 (59.0–72.6) | 0.76 (0.71–0.81) |
| *(155/228)* | 4+ symptom | 93.5 (88.5–96.5) | 37.0 (26.8–48.5) | 92.5 (87.0–95.5) | 41.1 (30.5–52.6) | 0.77 (0.70–0.84) |

N = total number of cases and controls, n = number of TBpos cases as defined by the MRS. Smear-positive (n = 289) and smear-negative (n = 102) participants were each assessed against all TBneg (n = 374) participants, giving a total of 663 and 476 included participants respectively.

participants a CRP cutoff-point of 6mg/L gave a sensitivity of 78.5% (95%CI 73.2–82.9%) and a specificity of 64.4% (95%CI 58.4–70.1%), whilst in HIV+ participants a cutoff-point of 22.7mg/L gave a sensitivity of 85.6% (95%CI 77.9–90.9%) and a specificity of 69.6% (95%CI 60.1–77.7%)). Sub-analyses by country of origin are reported in the supporting information, along with analyses using the XRS (S2 and S4 Tables). Sensitivity and specificity showed a large degree of heterogeneity between study sites, although the number of participants at some sites were small.

### CRP triage algorithm

A realistic implementation scenario for CRP triage testing would involve an algorithm in which a positive CRP triage result led to a confirmatory Xpert MTB/Rif test. Given such a scenario, the number of required confirmatory tests would be decreased by 40.6% using a CRP cutoff-point of 8mg/L. However, 55/221 TBpos participants with determinate Xpert results (24.9%) would be missed. By contrast, if Xpert MTB/Rif were implemented for all without CRP triage, 38 (17.2%) participants would be missed. Further details are provided in S6 Table.

### Discussion

Currently, CRP is the only available inexpensive POC-compatible investigation with the potential to be a suitable triage test for tuberculosis. This study for the first time provides an assessment of the performance of a quantitative laboratory-based CRP test for the diagnosis of presumed TB in patients presenting across five high-burden TB countries independent of HIV status [14, 15]. Whilst the specificity against culture-defined TB was close to 70%, sensitivity was sub-optimal at 77.7% at a cutoff-point of 10mg/L. For Xpert-defined TB CRP sensitivity was 90.7% and thus met the sensitivity as defined in WHO Triage-TPP but fell 13 percentage points short of the specificity of the TPP at 8mg/L. Cutoff-points of 8mg/L and 10mg/L were selected on the basis of previous work [10, 11] but the optimal cutoff-point seems to occur at higher CRP concentrations. Considering the most likely implementation scenario with Xpert as a confirmatory test (as indicated in the TPP), 18mg/L would be the best cutoff-point in our dataset for a triage test. Whilst the use of CRP as a triage test would reduce usage of confirmatory Xpert testing by about 40% at a cutoff-point of 8mg/L, the results suggest it would lead to approximately a quarter of culture-positive cases being missed, 8 percentage points more in absolute terms than with an Xpert-for-all strategy.

CRP concentrations tended to be higher in individuals with more symptoms at presentation and those with smear-positive disease, and it performed with greater sensitivity in these patient groups. In addition, higher CRP at presentation was associated with a shorter time to culture positivity, confirming previous findings [11]. As smear-positive TB cases likely contribute as much as 80% of total disease transmission [18], and as an effective POC triage test would likely lead to the consideration of TB in more patients, it is possible that despite suboptimal diagnostic performance, CRP could still contribute to the reduction of TB disease burden compared with current practice and to the rational use of Xpert cartridges. However, the suboptimal specificity might make the implementation of the test in an Xpert-algorithm challenging, especially in low prevalence settings. An algorithm that considers clinical risk factors in addition, or sequential testing [19], could be considered.

An algorithmic approach using CRP to triage individuals to confirmatory Xpert testing shows promise. Over half of culture-positive cases that would have been missed by this algorithmic approach were missed Xpert diagnoses, not a result of CRP test performance (S1 Fig). Of note, Xpert MTB/Rif was 82.8% sensitive for culture-positive TB in this study, lower than reported in meta-analyses [20, 21]. Xpert Ultra is likely to become the predominant TB test on

the Xpert platform and is shown to have improved sensitivity over Xpert MTB/Rif [22]. The transition to Ultra could lead to algorithm sensitivity gains, although would likely result in some decrease in specificity. Reducing Xpert cartridge consumption using an algorithmic approach would have important financial implications in low-income settings. Furthermore, in conjunction with the advent of smaller, more portable and robust Xpert platforms designed for use in LMIC community settings [23, 24], POC CRP testing in an algorithmic approach may provide an opportunity to increase test coverage at the location that the majority of TB patients initially access care [4], and could increase diagnostic pick-up. However, this study used laboratory-based methods to quantify CRP and the performance of any algorithm would need to be confirmed using POC-testing platforms to assess suitability for the field. Performance of field-based POC clinical tests vary widely, and so this would not be an insignificant translational barrier.

Although CRP did not meet the TPP overall, use of alternate cutoff-points based on HIV sero-status allowed for improved performance in HIV+ participants. This suggests that CRP at one defined cutoff-point might not meet the challenge of being a catch-all triage test for the global TB epidemic, but different cutoff-points might be necessary in high and low HIV-prevalence settings. Whilst CRP had excellent sensitivity in HIV+ participants at the pre-defined cutoff-points, specificity was far lower than in HIV- participants. The latter is likely to reflect one or both of the increased prevalence of infective pathologies or the baseline CRP found in HIV+ patients [25–28]. Using a higher CRP cutoff-point for HIV+ participants improved specificity, almost meeting TPP using the XRS. Recent studies have suggested CRP has a role in TB screening of HIV+ individuals [11, 12]. This study supports previous work suggesting it could also have a role in triaging HIV+ patients undergoing evaluation for presumed TB, with CRP cutoff-points adjusted accordingly [14, 15]. Indeed, the WHO advocates HIV testing whenever TB is presumed [29] and mandatory POC HIV testing to inform CRP cutoff-point adjustment could assist in further operationalising HIV screening within TB programs. Despite concerns that the diagnostic specificity in HIV- participants would be compromised by the burden of alternative pyogenic and inflammatory conditions, the specificity of CRP for culture-positive TB in HIV negative participants was satisfactory. The suboptimal sensitivity seen in this patient group may be due to low levels of systemic immune activation at lower bacillary burdens, as suggested by the poor performance of CRP in smear negative disease.

CRP performance varied by study site (in a post-hoc analysis reported in the supporting information), tending to perform better outside south-east Asia. Previous data assessing the role of CRP in the diagnosis of TB have largely been from sub-Saharan Africa [10]. CRP has been found to better distinguish bacterial from viral illness in febrile patients in Asia compared to those from equatorial Africa. This is likely because of the burden of malaria and parasitic co-infection in the latter [30, 31]. However, it is likely the burden of TB/bacterial co-infection is higher in South-East Asia [32], adversely impacting the diagnostic performance of CRP in these settings. Furthermore, previous work by Brown et al. [33] suggests there may be intrinsic differences in host CRP response to TB between different ethnicities and mycobacterial strain type, raising the prospect of adjustment of CRP cutoff-points by geography in order to improve diagnostic performance. More work is required to better understand these observations.

This study has a number of strengths. It is the first study to explore the utility of CRP as a triage test and reports the diagnostic characteristics of CRP at a range of cutoff-points. It used participants from a range of study locations and with a range of mycobacterial burdens, and in doing so is largely representative of the global TB epidemic. The use of a laboratory-based analyser allowed assessment of the sensitivity and specificity of CRP without concerns introduced by the accuracy of POC platforms.

However, it also has some limitations. Smear negative participants made up 26% of TB cases in this study compared to approximately 50% generally reported in clinical practice [34]. CRP performed less well in these participants, and so overall test performance in a prospective setting may be worse than reported here. Facilities enabling liquid sputum culture are often not available at peripheral health centres in high-burden countries and so its use as a confirmatory test may be limited in these settings at present. However, Xpert could act as such a test; it is being rapidly scaled-up in high burden countries and endorsed by the WHO as a follow-on test for smear-negative PTB [35]. Whilst this study used laboratory-based CRP measurements, triage testing in peripheral health centres would require a POC platform. Performance of such platforms vary widely and would require further evaluation against a laboratory standard. Furthermore, the performance of the proposed algorithm would need to be determined prospectively using POC platforms in a population with more co-morbidities before any implementation of CRP for triage could occur.

CRP may not be the catch-all triage test that the global TB epidemic requires. However, it shows promise in certain patient groups: this study suggests a possible role in HIV+ individuals undergoing evaluation for presumed TB. Further research is required in a prospective study using POC-platforms to more thoroughly evaluate its capabilities in the field. Should these studies prove its utility, and in conjunction with the ongoing development of community-based Xpert platforms, it raises the possibility of increasing access to TB testing in resource-poor settings.

## Supporting information

**S1 Fig. Participant flow diagram for CRP cutoff-point of 8mg/L.**
(TIF)

**S2 Fig. Venn diagram showing distribution of TB-related symptoms in TBpos participants on presentation to health care (n = 391).** Numbers represent number of participants presenting with the indicated symptom complex. Total TBpos participants presenting with night sweats n = 227, with haemoptysis n = 88, with recent weight loss n = 269 and with none of the above n = 61.
(TIF)

**S3 Fig. Scatterplot of CRP concentration against time to sputum liquid culture positivity for TBpos participants.** Blue dots represent individual CRP data points, red line demonstrates line of best fit and green demonstrates LOWESS locally weighted smoothing. As CRP concentration decreased linearly with increased time to sputum culture positivity, Pearson's correlation was applied and found to be significant with a coefficient of -0.234 (p = 0.003).
(TIF)

**S4 Fig.** ROC curve analysis of CRP performance in the diagnosis of tuberculosis using A) the MRS and B) the XRS, by HIV status. The shaded orange area represents the sensitivity and specificity combinations that meet at least the minimum target of one of the Target Product Profile characteristics. Minimum (Sens 90% Spec 70%) and Optimal (Sens 95% Spec 80%) Target Product Profile targets are plotted as a black diamond and black triangle respectively. Predefined CRP cutoff-points are plotted as red circles and yellow squares with optimal CRP cutoff-points plotted as green diamonds. In A) optimal CRP cutoff-points occurred at 23mg/L for HIV+ participants and 6mg/L for HIV- participants. In B), these cutoff-points were 29mg/L (Sensitivity 90.3% Specificity 68.3%) and 11mg/L (Sensitivity 86.0% and Specificity 68.8%) respectively. AUC = Area Under Curve; ROC = Receiver Operating Characteristic; CRP = C-reactive protein; MRS = Microbiological reference standard; XRS+ Xpert MTB/Rif reference

standard.
(TIF)

**S5 Fig.** ROC curve analysis of CRP performance in the diagnosis of tuberculosis against the MRS by A) Age of serum sample at time of CRP analysis and B) without Vietnamese sera older than 3,000 days. Variation by study site prompted post-hoc exploratory analysis, which revealed CRP diagnostic performance was significantly worse in pre-2011 samples taken in Viet Nam compared to those taken later. A significant negative correlation between CRP levels and serum sample age was seen in Vietnamese TBpos participants, without a similar correlation being observed in TBneg samples. Optimal sensitivity and specificity against MRS excluding this pre-2011 data (B) was 84.5% (95%CI 79.7–88.3) and 69.5% (95%CI 64.5–74.0) respectively at 12mg/L cutoff-point. This also explains much of the reported difference in CRP performance between the XRS and the MRS, as the pre-2011 Vietnamese participants did not receive Xpert testing. The shaded orange area represents the sensitivity and specificity combinations that meet at least the minimum target of one of the Target Product Profile characteristics. Minimum (Sens 90% Spec 70%) and Optimal (Sens 95% Spec 80%) Target Product Profile targets are plotted as a black diamond and a black triangle respectively. Pre-defined CRP cutoff-points are plotted as red circles (8mg/L) and yellow squares (10mg/L) with optimal CRP cutoff-points plotted as green diamonds. AUC = Area Under Curve; ROC = Receiver Operating Characteristic; CRP = C-reactive protein; MRS = Microbiological Reference Standard.
(TIF)

**S1 Table. Characteristics of participants who received Xpert MTB/Rif testing.**
(PDF)

**S2 Table. Sensitivity and specificity of CRP against the XRS by study site, smear status, HIV status and number of symptoms at presentation.** N = total number of cases and controls, n = number of TBpos cases as defined by the XRS. Smear-positive (n = 289) and smear-negative (n = 102) participants were each assessed against all TBneg (n = 374) participants, giving a total of 663 and 476 included participants respectively.
(PDF)

**S3 Table. Threshold analysis of the performance of CRP against the XRS using ROC curve data, overall and by HIV status.**
(PDF)

**S4 Table Sensitivity and specificity of CRP against the MRS by study site N = total number of cases and controls, n = number of TBpos cases as defined by the MRS.**
(PDF)

**S5 Table. Threshold analysis of the performance of CRP against the MRS using ROC curve data, overall and by HIV status.**
(PDF)

**S6 Table. The diagnostic accuracy of CRP triage against the MRS at varying CRP triage thresholds.** In this algorithm, a positive CRP triage test above the stated threshold would trigger patients to move forward to confirmatory testing by the stated method. The performance of chest X-ray (CXR) as a triage test was assessed for comparison. A CXR was defined as positive for TB if the radiographic appearance was judged to be consistent with typical or atypical TB. The sensitivity and specificity of a single Xpert MTB/Rif against the MRS is also presented.
(PDF)

**S7 Table. Univariate linear regression analysis of the relationship between CRP and selected variables.** All variables with a p value <0.1 were considered for inclusion into multi-variable analysis. Akaike Information Criterion values were then used to build the best model. CD4 count was excluded from the multivariable model despite meeting significance as only 95 study participants had recorded values. AIC was not reported for CD4 count as sample size differed from other variables.
(PDF)

**S1 File. Study protocol.**
(DOCX)

**S1 Dataset.**
(XLSX)

## Acknowledgments

The authors thank Anna Mantsoki, Audrey Albertini, Marta Fernandez Suarez and Ranald Sutherland for helping with the conceptualization of this work, the participants who provided the samples and the clinical and laboratory teams at the partner sites for their efforts in performing the study.

## Author Contributions

**Conceptualization:** Samuel G. Schumacher, Claudia M. Denkinger.

**Data curation:** Romain Wyss.

**Formal analysis:** Thomas H. A. Samuels.

**Funding acquisition:** Claudia M. Denkinger.

**Investigation:** Romain Wyss.

**Methodology:** Romain Wyss, Claudia M. Denkinger.

**Supervision:** Samuel G. Schumacher, Claudia M. Denkinger.

**Writing – original draft:** Thomas H. A. Samuels.

**Writing – review & editing:** Stefano Ongarello, David A. J. Moore, Samuel G. Schumacher, Claudia M. Denkinger.

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
