## [Decision Letter · Decision Letter 0]

5 Aug 2020

PONE-D-20-19334

Evaluation of the diagnostic performance of laboratory-based c-reactive protein as a triage test for active pulmonary tuberculosis

PLOS ONE

Dear Dr. Schumacher,

Thank you for submitting your manuscript to PLOS ONE. After careful consideration, we feel that it has merit but does not fully meet PLOS ONE’s publication criteria as it currently stands. Therefore, we invite you to submit a revised version of the manuscript that addresses the points raised during the review process.

Please submit your revised manuscript. If you will need more time than this to complete your revisions, please reply to this message or contact the journal office at plosone@plos.org. Please include the following items when submitting your revised manuscript:

We look forward to receiving your revised manuscript.

Kind regards,

Frederick Quinn

Academic Editor

PLOS ONE

Journal Requirements:

Reviewers' comments:

Reviewer's Responses to Questions

**Comments to the Author**

1. Is the manuscript technically sound, and do the data support the conclusions?

Reviewer #1: Yes

Reviewer #2: Yes

Reviewer #3: Yes

2. Has the statistical analysis been performed appropriately and rigorously? 

Reviewer #1: No

Reviewer #2: Yes

Reviewer #3: Yes

3. Have the authors made all data underlying the findings in their manuscript fully available?

Reviewer #1: Yes

Reviewer #2: Yes

Reviewer #3: Yes

4. Is the manuscript presented in an intelligible fashion and written in standard English?

Reviewer #1: Yes

Reviewer #2: Yes

Reviewer #3: Yes

5. Review Comments to the Author

Reviewer #1: This study is a retrospective matched case-control study of the diagnostic accuracy of C-reactive protein for the diagnosis of pulmonary tuberculosis. The strengths include the size and diversity of the population tested. However I have concerns that several of the analyses are inappropriate for the case-control design of this study.

Major points

Lines 110-119: How were controls selected? Were controls unmatched? From line 159 it seems they were matched on HIV status.

Lines 175-177 - I don't think it's appropriate to conduct a linear regression on case-control data without correcting for selection bias associated with oversampling of cases. For example, see doi: 10.1093/biostatistics/kxt041

Additionally, with a data-driven variable selection procedure for your model, please also include stability / sensitivity analyses. See doi:10.1002/bimj.201700067

Lines 171-172: Similarly, the ROC analysis should be adjusted for the case-control design. See doi: 10.1016/j.acra.2013.03.004, and doi: 10.1373/clinchem.2012.186007

Minor points

line 29 "no study has assessed its suitability as a triage test." Similarly stated on line 353. What about doi: 10.1097/QAD.0000000000001902, and doi: 10.5588/ijtld.13.0519 ?

Lines 83-88 citation?

Line 165 Parametric is a general term for any probability distribution. Do you mean normally distributed? How was this determined?

Table 1: Please include Xpert status for cases

Line 361 "50% generally reported in clinical practice" This needs a citation. Maybe in HIV + cohorts, but generally?

Reviewer #2: The manuscript is well written, the study design and the analysis are clear and the analysis has considered a range of interesting secondary questions; I have a few comments on the analysis and discussion.

Your paper makes it clear that you are concerned with CRP in conjunction with another test, which you expect will usually be Xpert MTB/Rif (but could be culture). So in referring to the TPP specified minimum sensitivity and specificity, should you not be looking at the figures for the composite of CRP followed (if threshold is met) by Xpert, and not CRP alone? If one can choose the appropriate CRP threshold to obtain high enough sensitivity, and then follow it with Xpert, why should we demand high specificity of CRP alone? Your table S6 notes that with Xpert as the second part of this composite CRP-Xpert testing one sees 100% specificity.

You report that CRP fails to meet the TPP specified requirement of at least 70% specificity but in this paper you are not suggesting CRP alone. Even a CRP specificity less than 70% could result in some reduction of Xpert testing (the only benefit of high CRP specificity in your composite testing model seems to be a health system benefit - as opposed to improved diagnostic yield which is not diminished by the low specificity of CRP, when used with Xpert or culture).

Line 37-38: The abstract states that "The PTB reference standard was a contemporaneously collected single sputum MGIT culture result."

Line 104-106: Later in Materials and Methods it is stated that "To be included in the analysis all participants had to provide both a blood sample and at least 2 sputum samples at enrolment prior to initiation of anti-tuberculous therapy."

Is the reference standard for diagnosis of TB one or two MGIT TB cultures? If two, are they performed on different specimens?

Line 319-20: "However, the suboptimal specificity might make the implementation of the test in an Xpert-algorithms impossible." The word 'algorithm' should be singular (or the word 'an' should be removed if the plural is intended).

In your discussion of the sensitivity of CRP among HIV+ and HIV- persons with TB you note the use of different threshold values for HIV+ persons, or the possible addition of clinical features to an algorithm. No mention is made of the possibility of more than one CRP measurement per patient during the diagnostic evaluation period. Your analysis and others make it clear that CRP is more likely to be elevated when there are clinical (eg., fever) and pathological (eg., hemoptysis, indicative of cavitation) features of intense inflammation, or its sequelae. It is likely that symptoms become more obvious the longer one has untreated TB, but sequential changes in CRP among persons suspected of having TB have not, as far as I am aware, been examined. Work by Stephen Lawn and others has looked at sequential CRP in response to TB treatment which may offer a diagnostic confirmation when a fall in CRP is observed.

Reviewer #3: This is a well written manuscript reporting the diagnostic accuracy of CRP as a TB triage test among patients with at least 1 TB symptom. The strengths of this manuscript include: 1) well-characterized cohorts, 2) geographic diversity (5 TB endemic countries) and 3) robust microbiologic and combined micro/clinical reference standards. The authors found that among patients with any TB symptom, CRP did not meet min TPP criteria for TB triage testing (insufficient sensitivity and specificity) at 8 or 10 mg/L. However, my main concern with this conclusion is that TB triage testing would not be applied to individuals with any TB symptom, unless they were at high risk for TB (e.g., HIV-positive). For everyone else, TB triage testing would be applied to patients with cough ≥2 weeks, however CRP diagnostic accuracy data for this group was not presented in this analysis and is likely to yield different diagnostic accuracy results. If symptom duration data is available, diagnostic accuracy of CRP should be assessed in the appropriate target populations for triage testing (cough ≥2 weeks if HIV-neg, any TB symptom in the past 1 month if HIV-pos).

Other major comments:

1. TB test nomenclature - Because the TB literature has used 'triage test' to refer to both primary screening tests (a test applied to an unselected group to identify individuals with suspected disease requiring confirmatory testing) as well as secondary screening tests which is the conventional definition (a test applied to individuals with suspected to disease with the goal of reducing the proportion requiring confirmatory testing), it would be helpful for the authors to be very explicit with how this manuscript defines triage test.

-- Abstract - please consider clarifying definition of triage test here. Currently, the first sentence could also be used to describe a primary screening test.

- Intro, 2nd paragraph - I'm not familiar with the term 'two-stage diagnostic algorithm,' used here and find it a bit confusing. Seems like 'two-stage diagnostic algorithm' doesn't include cough ≥2 weeks (screening test) but describes triage testing and confirmatory testing when applied to patients with cough ≥ 2 weeks. Please consider using conventional terminology/nomenclature and clearly defining these when more than 1 definition is commonly used.

2. Geographic differences in CRP diagnostic accuracy - CRP sensitivity was lower in Asian countries (Cambodia, Vietnam). It would be helpful to discuss this a bit more in the discussion. Marc Lipman (https://bmcinfectdis.biomedcentral.com/articles/10.1186/s12879-016-1612-1) reported similar findings: Asian ethnicity was associated with lower median CRP levels among those with TB.

-- Tables S2 and S4 - why is CRP diagnostic accuracy for Vietnam so different in Table S2 and S4?

Minor comments:

Intro:

1. Page 4, line 78. Should change "PTB outpatients" to "outpatients with suspected TB (or outpatients with TB symptoms)."

2. Page 5, lines 82-83. Are prior studies limited when the reference cited is a meta-analysis that includes 6 studies that included HIV-negative patients? Also, please consider briefly (in 1 or 2 sentences) summarizing the results of this SR/MA that are relevant to this manuscript, which would provide readers with some context for the following 2 sentences.

3. Page 5, lines 83-85. This sentence suggests that CRP specificity will be lower among HIV-negative patients, is that the intent? I would assume the opposite because of lower rates of infection relative to HIV-positive patients.

4. Page 5, lines 85-86. "Individuals who present to healthcare with symptoms suggestive of PTB (therefore requiring triage instead of screening)..." With the current TB testing algorithm, patients with TB symptoms require confirmatory testing. I understand the point being made is that CRP triage testing can be expected to have reduced specificity relative to CRP screening but I think the comment in parentheses obscures this point.

5. Should the last reference in paragraph 3 (ref 13) be deleted here?

Methods:

1. Page 6, lines 99-101. Please provide more detail about the study population (e.g., outpatients, inpatients or both?)

2. Page 6, lines 101-104. Same as above major comment: If symptom duration data are available, please pull out cough ≥2 weeks and any 1/4 TB symptom in past month.

3. Last sentence of introduction mentions that a high sensitivity CRP (hsCRP) assay was used. The methods section does not mention this in the Index test section. Prior studies of CRP in TB use a standard sensitivity assay. Can the authors explain why a high sensitivity assay was chosen and whether the type of assay (hs or standard) would impact results?

Results:

1. Table 1 - TB symptoms at presentation. Same as 2nd Methods comment. Also, if presence of at least 1 TB symptom was a requirement for this analysis and no patients had 0 symptoms, shouldn't the number of symptoms at presentation begin with 1 (opposed to '0-1')?

2. Page 14, Sensitivity and Specificity analysis - same as above major comment.

3. Page 17, 2nd paragraph - how were optimal cut-points determine? by AUROC? If so, this approach weighs sensitivity and specificity equally, but clinically, most ppl would feel that 1 missed case of active TB far outweighs a 1 false-positive case. From this clinical perspective, it might make more sense to fix sensitivity at 90% to identify optimal CRP cut-point. FYI: the manuscript also mentions alternative CRP cut-points in the Discussion (line 288-290, 324 paragraph).

4. Page 17, 3rd paragraph - related to major comment above. Recommend using CRP triage testing as it would be applied in currently recommended TB testing algorithms (e.g., HIV negative patients with cough≥ 2 weeks)

Discussion:

1. Page 19, line 298 - replace 'conversion' with 'positivity'?

2. Page 19, lines 302-303 - this sentence is unclear.

6. PLOS authors have the option to publish the peer review history of their article (what does this mean?). If published, this will include your full peer review and any attached files.

Reviewer #1: No

Reviewer #2: **Yes: **Richard A Bedell

Reviewer #3: No

---

## [Author Response · Author response to Decision Letter 0]

10 Mar 2021

For ease of reference, the specific reviewer and editor comments and our associated replies are detailed in a table format in the document uploaded as a response to the reviewers. For ease of access, these are listed here too. 

We note the editor’s request to review our document formatting and are glad to say that we have amended our title page to better fit editorial requirements. We apologise this was overlooked in our initial submission. On review of the rest of the document, we believe all other formatting requirements have been followed.

We will address reviewers points below sequentially. 

Reviewer 1:

Major points: 

Comment

Line 110-119: How were controls selected? Were controls unmatched? From line 159 it seems they were matched on HIV status.

Response:

This point has been addressed by further clarifying how cases and controls were selected and the fact that they were matched by HIV status in the relevant section of the methods. 

Comment:

Line 175-177: I don't think it's appropriate to conduct a linear regression on case-control data without correcting for selection bias associated with oversampling of cases. For example, see doi: 10.1093/biostatistics/kxt041

Response: 

We believe that this methodology proposed by the reviewer is designed to address circumstances in which cases and controls are selected with unequal probability and therefore we are not sure that it applies to our study. 

Comment:

Additionally, with a data-driven variable selection procedure for your model, please also include stability / sensitivity analyses. See doi:10.1002/bimj.201700067

Response:

We agree with the reviewer that in data-driven variable selection a sensitivity analysis is an important part of best-practice methodology. However, as the number of variables we included a priori in the univariate part of our analysis was small (8 in total), we felt that carrying out a full sensitivity analysis to justify our variable selection wasn't necessary.

Comment:

Line 171-172: Similarly, the ROC analysis should be adjusted for the case-control design. See doi: 10.1016/j.acra.2013.03.004, and doi: 10.1373/clinchem.2012.186007

Response:

As with the point above on our linear regression, we believe that this methodology proposed by the reviewer is designed to address circumstances in which cases and controls are selected with unequal probability and therefore we are not sure that it applies to our study. 

Minor points:

Comment:

Line 29: no study has assessed its suitability as a triage test." Similarly stated on line 353. What about doi: 10.1097/QAD.0000000000001902, and doi: 10.5588/ijtld.13.0519 ?

Response: 

We agree with the reviewer that the referenced work has investigated CRP as a triage test in HIV+ participants but that there isn’t a dedicated study we are aware of that looks at HIV-negative patients. We have amended the manuscript to reflect this. 

Comment:

Line 83-89: Citation

Response:

We agree with the reviewer and have actioned this point

Comment:

Line 165: Parametric is a general term for any probability distribution. Do you mean normally distributed? How was this determined?

Response:

We agree with the reviewer and have changed this to ‘normally-distributed’ in the manuscript

Comment:

Table 1: Please include Xpert status for cases

Response:

We agree with the reviewer and have actioned this point

Comment:

Line 361: 50% generally reported in clinical practice" This needs a citation. Maybe in HIV + cohorts, but generally?

Response:

We agree with the reviewer and have cited this point

Reviewer 2:

Major points:

Comment:

Your paper makes it clear that you are concerned with CRP in conjunction with another test, which you expect will usually be Xpert MTB/Rif (but could be culture). So in referring to the TPP specified minimum sensitivity and specificity, should you not be looking at the figures for the composite of CRP followed (if threshold is met) by Xpert, and not CRP alone? If one can choose the appropriate CRP threshold to obtain high enough sensitivity, and then follow it with Xpert, why should we demand high specificity of CRP alone? Your table S6 notes that with Xpert as the second part of this composite CRP-Xpert testing one sees 100% specificity.

You report that CRP fails to meet the TPP specified requirement of at least 70% specificity but in this paper you are not suggesting CRP alone. Even a CRP specificity less than 70% could result in some reduction of Xpert testing (the only benefit of high CRP specificity in your composite testing model seems to be a health system benefit - as opposed to improved diagnostic yield which is not diminished by the low specificity of CRP, when used with Xpert or culture).

Response:

We agree with the reviewers point that the overall performance of a two-step algorithm is highly important to consider. However, our work is designed to focus specifically on the question of performance related to first step of that algorithm and, as the reviewer states in the second paragraph, its health systems benefit by rationalising the use of confirmatory testing. The WHO target product profile for a triage test is designed to be in reference to the confirmatory test chosen and so is a good standard for our research question. The 70% specificity target for a triage test in this instance accounts for the fact that at lower prevalence settings a lesser specificity would cause the number of false positives to exceed the number of true positives. This would erode the health system benefit to the point that they may be outweighed by the potential loss of trust in the triage test. We have altered the manuscript to try and make this aim clearer. 

Comment:

In your discussion of the sensitivity of CRP among HIV+ and HIV- persons with TB you note the use of different threshold values for HIV+ persons, or the possible addition of clinical features to an algorithm. No mention is made of the possibility of more than one CRP measurement per patient during the diagnostic evaluation period. Your analysis and others make it clear that CRP is more likely to be elevated when there are clinical (eg., fever) and pathological (eg., hemoptysis, indicative of cavitation) features of intense inflammation, or its sequelae. It is likely that symptoms become more obvious the longer one has untreated TB, but sequential changes in CRP among persons suspected of having TB have not, as far as I am aware, been examined. Work by Stephen Lawn and others has looked at sequential CRP in response to TB treatment which may offer a diagnostic confirmation when a fall in CRP is observed.

Response:

We agree with the reviewer on this point that this represents a very interesting line of enquiry and one that should be pursued. Unfortunately, we do not have the data to be able to pursue it in this work. However, we have altered the discussion to make mention of it.

Minor points:

Comment:

Line 37-38: The abstract states that "The PTB reference standard was a contemporaneously collected single sputum MGIT culture result."

Line 104-106: Later in Materials and Methods it is stated that "To be included in the analysis all participants had to provide both a blood sample and at least 2 sputum samples at enrolment prior to initiation of anti-tuberculous therapy."

Is the reference standard for diagnosis of TB one or two MGIT TB cultures? If two, are they performed on different specimens

Response:

We agree with the reviewers point that these statements in the manuscript are somewhat and unintentionally contradictory. We have amended the abstract to reflect the methods – the reference standard is two MGIT as is now stated in the Methods. 

Comment:

Line 311: However, the suboptimal specificity might make the implementation of the test in an Xpert-algorithms impossible." The word 'algorithm' should be singular (or the word 'an' should be removed if the plural is intended).

Response:

We agree with the reviewer and have actioned this point

Reviewer 3

Major points:

Comment:

However, my main concern with this conclusion is that TB triage testing would not be applied to individuals with any TB symptom, unless they were at high risk for TB (e.g., HIV-positive). For everyone else, TB triage testing would be applied to patients with cough ≥2 weeks, however CRP diagnostic accuracy data for this group was not presented in this analysis and is likely to yield different diagnostic accuracy results. If symptom duration data is available, diagnostic accuracy of CRP should be assessed in the appropriate target populations for triage testing (cough ≥2 weeks if HIV-neg, any TB symptom in the past 1 month if HIV-pos).

Response: 

We agree with the reviewer that triage testing would usually be offered to the patient population they specify. However, in this work we tried to cast a wider net for our primary analysis and chose current cough as opposed to cough ≥2 weeks as this was shown to be a more sensitive screening algorithm for PTB in the following paper: https://www.ncbi.nlm.nih.gov/pmc/articles/PMC3391193/. A sub-analysis to assess the diagnostic performance of CRP in the target populations specified by the reviewer would be a good addition to our manuscript but unfortunately, we do not have sufficient granularity in our symptom data to do so.

Comment:

TB test nomenclature - Because the TB literature has used 'triage test' to refer to both primary screening tests (a test applied to an unselected group to identify individuals with suspected disease requiring confirmatory testing) as well as secondary screening tests which is the conventional definition (a test applied to individuals with suspected to disease with the goal of reducing the proportion requiring confirmatory testing), it would be helpful for the authors to be very explicit with how this manuscript defines triage test.

-- Abstract - please consider clarifying definition of triage test here. Currently, the first sentence could also be used to describe a primary screening test.

- Intro, 2nd paragraph - I'm not familiar with the term 'two-stage diagnostic algorithm,' used here and find it a bit confusing. Seems like 'two-stage diagnostic algorithm' doesn't include cough ≥2 weeks (screening test) but describes triage testing and confirmatory testing when applied to patients with cough ≥ 2 weeks. Please consider using conventional terminology/nomenclature and clearly defining these when more than 1 definition is commonly used.

Response:

We agree with the reviewer that our use of ‘triage test’ and ‘two-stage diagnostic algorithm’ in the manuscript is not specific enough as it stands and we have altered the relevant sections in the abstract and main text to make them so, in line with their suggestion.

Comment:

Geographic differences in CRP diagnostic accuracy - CRP sensitivity was lower in Asian countries (Cambodia, Vietnam). It would be helpful to discuss this a bit more in the discussion. Marc Lipman (https://bmcinfectdis.biomedcentral.com/articles/10.1186/s12879-016-1612-1) reported similar findings: Asian ethnicity was associated with lower median CRP levels among those with TB. -- Tables S2 and S4 - why is CRP diagnostic accuracy for Vietnam so different in Table S2 and S4?

Response:

We agree with the reviewer’s point here and have expanded this paragraph in the discussion to include the suggested reference. Sadly, we do not have strain data for our patients to explore this line of enquiry further. Regarding the reviewers point about the difference between Vietnamese data in Table S2 and S4 – Xpert MTB/Rif testing was carried out only on a more recent subgroup of the total Vietnamese cohort – this data is presented in S2. Culture data for the total Vietnamese cohort is presented in S4. We did further analysis to interrogate this difference. Culture performance in the subgroup presented in S2 was better than for the cohort as a whole in S4, although the difference was less marked that presented between S2 and S4. We conducted some analysis as to the effect of age of sample in the cohort as a whole and found no significant correlation between age and CRP levels. However, there was a signal when this analysis was applied to the Vietnamese data alone. Against a sample-degradation hypothesis, CRP has been shown to be stable in serum for extended periods of time DOI: 10.1016/j.clinbiochem.2013.12.014. We feel it is possible there is some age-related effect here, but that also the overall performance of CRP in Vietnam is also related to host factors and strain time as laid out in the paper the reviewer mentions.

Minor points:

Comment:

Line 78: Should change "PTB outpatients" to "outpatients with suspected TB (or outpatients with TB symptoms)."

Response:

We agree with the reviewer and have actioned this point

Comment:

Line 82-83: Are prior studies limited when the reference cited is a meta-analysis that includes 6 studies that included HIV-negative patients? Also, please consider briefly (in 1 or 2 sentences) summarizing the results of this SR/MA that are relevant to this manuscript, which would provide readers with some context for the following 2 sentences.

Response:

We agree with the reviewer and have actioned this point, changing the wording of the relevant part of the manuscript and summarising the results of this SR. 

Comment:

Line 83-85: This sentence suggests that CRP specificity will be lower among HIV-negative patients, is that the intent? I would assume the opposite because of lower rates of infection relative to HIV-positive patients.

Response:

We agree with the reviewer that this section as it stands didn’t follow. We have amended it accordingly. 

Comment:

Line 85-86: Individuals who present to healthcare with symptoms suggestive of PTB (therefore requiring triage instead of screening)..." With the current TB testing algorithm, patients with TB symptoms require confirmatory testing. I understand the point being made is that CRP triage testing can be expected to have reduced specificity relative to CRP screening but I think the comment in parentheses obscures this point

Response:

We agree with the reviewer and have actioned this point to make it clearer. 

Comment:

Line 90: Should the last reference in paragraph 3 (ref 13) be deleted here?

Response:

We agree with the reviewer and have actioned this point

Comment:

Line 99-101: Please provide more detail about the study population (e.g., outpatients, inpatients or both?)

Response: 

We agree with the reviewer that more detail is required and have amended the manuscript

Comment:

Line 101-104: Same as above major comment: If symptom duration data are available, please pull out cough ≥2 weeks and any 1/4 TB symptom in past month.

Response:

As we have mentioned above in response to the major comment, whilst we agree with the reviewer that this would be a worthy analysis, we do not have the data to complete it.

Comment:

Last sentence of introduction mentions that a high sensitivity CRP (hsCRP) assay was used. The methods section does not mention this in the Index test section. Prior studies of CRP in TB use a standard sensitivity assay. Can the authors explain why a high sensitivity assay was chosen and whether the type of assay (hs or standard) would impact results.

Response:

We agree with the reviewers point that there is a discrepancy here. The analyser employed in this study does have the ability to process hsCRP. However, for this study we employed a standard sensitivity mode. The manuscript has been amended to reflect this and resolve the discrepancy

Comment:

Table 1: TB symptoms at presentation. Same as 2nd Methods comment. Also, if presence of at least 1 TB symptom was a requirement for this analysis and no patients had 0 symptoms, shouldn't the number of symptoms at presentation begin with 1 (opposed to '0-1')?

Response:

We agree with the reviewers point here – the 0-1 symptoms at presentation were a mistake – this should be 1 symptom. We have amended the manuscript and figures accordingly

Comment:

Page 17, 2nd paragraph: how were optimal cut-points determine? by AUROC? If so, this approach weighs sensitivity and specificity equally, but clinically, most ppl would feel that 1 missed case of active TB far outweighs a 1 false-positive case. From this clinical perspective, it might make more sense to fix sensitivity at 90% to identify optimal CRP cut-point. FYI: the manuscript also mentions alternative CRP cut-points in the Discussion (line 288-290, 324 paragraph).

Response: 

We agree with the reviewers point that further clarity is required as to the methodology employed to calculate optimal cutoff-points. We have now amended the methods section of the manuscript to make this clearer. Sensitivity and specificity were weighed equally in the TPP of the WHO and we feel that the importance of one versus the other depends on the TB prevalence in a given setting. As a result, we have gone with the methodology above to specify optimal cutoff-points as it lends weight to the impact of both aspects of test performance.

Comment:

Page 17, 3rd paragraph: related to major comment above. Recommend using CRP triage testing as it would be applied in currently recommended TB testing algorithms (e.g., HIV negative patients with cough≥ 2 weeks)

Response:

We have expanded on the reasoning behind our decision-making in regard to this point in the response to the major comment above. 

Comment:

Line 298: replace 'conversion' with 'positivity'?

Response:

We agree with the reviewer and have actioned this point

Comment:

Line 302-303: this sentence is unclear

Response:

We agree with the reviewer and have amended this sentence

---

## [Decision Letter · Decision Letter 1]

16 Apr 2021

PONE-D-20-19334R1

Evaluation of the diagnostic performance of laboratory-based c-reactive protein as a triage test for active pulmonary tuberculosis

PLOS ONE

Dear Dr. Schumacher,

Thank you for submitting your manuscript to PLOS ONE. After careful consideration, we feel that it has merit but does not fully meet PLOS ONE’s publication criteria as it currently stands. Therefore, we invite you to submit a revised version of the manuscript that addresses the points raised during the review process.

Please submit your revised manuscript. If you will need significantly more time to complete your revisions, please reply to this message or contact the journal office at plosone@plos.org. Please include the following items when submitting your revised manuscript:

We look forward to receiving your revised manuscript.

Kind regards,

Frederick Quinn

Academic Editor

PLOS ONE

Reviewers' comments:

Reviewer's Responses to Questions

**Comments to the Author**

1. If the authors have adequately addressed your comments raised in a previous round of review and you feel that this manuscript is now acceptable for publication, you may indicate that here to bypass the “Comments to the Author” section, enter your conflict of interest statement in the “Confidential to Editor” section, and submit your "Accept" recommendation.

Reviewer #1: (No Response)

Reviewer #2: (No Response)

Reviewer #4: (No Response)

2. Is the manuscript technically sound, and do the data support the conclusions?

Reviewer #1: Partly

Reviewer #2: Yes

Reviewer #4: Partly

3. Has the statistical analysis been performed appropriately and rigorously? 

Reviewer #1: I Don't Know

Reviewer #2: Yes

Reviewer #4: Yes

4. Have the authors made all data underlying the findings in their manuscript fully available?

Reviewer #1: Yes

Reviewer #2: Yes

Reviewer #4: Yes

5. Is the manuscript presented in an intelligible fashion and written in standard English?

Reviewer #1: Yes

Reviewer #2: Yes

Reviewer #4: Yes

6. Review Comments to the Author

Reviewer #1: 3 of the 4 major points I raised were dismissed without substantive discussion. I have asked to have the manuscript reviewed by a statistician.

Reviewer #2: Thank for addressing my previous points (and apologies for my misconstrual of the TPP specificity).

A few questions arise on rereading that you are likely able to address easily:

(1) In Table 1 / Smear status, we are told there were 289 participants S+C+ and 102 S-C+, whereas in Table 3 to the left of S+C+ we are given n=663 and for S-C+ we are given n=476; these numbers are not the numbers for S+C+ and S-C+ and must refer to [S+C+ plus all TB test neg] and [S-C+ plus all TB neg] but this is not immediately clear on first reading (or second reading I confess) - it might help to explicitly state in the line just below the table "Smear-positive and smear-negative participants were each assessed against all TBneg (n= 374)", or to insert another line in the table for All TB test neg.

(2) In the Discussion you note (in lines 306-309) that using a CRP cut off of 8 mg/L would result in approximately 26% of culture positive cases being missed. I expect this 26% to accord with the numbers given in Supplementary Figure 1 which relates to the use of the 8 mg/L cut off. In S Fig 1 it appears that a total of 79 culture + cases occurred among those with CRP <8 mg/L (with the total of all culture positive cases being 79 + 312 = 391) which looks like 20.2% (79/391) were missed (because CRP <8). Where does 26% come from? Is there a typographical error and you mean to continue referring to an 18 mg/L cut off, not an 8 mg/L cut off as the text currently states? (In that case it would obviously have nothing to do with S Fig 1.)

Reviewer #4: The study was generally well written. Please note that this is my first review of this paper. Although I have a long list of comments, they are mainly minor and for clarification. The only major comment is an error which can be easily corrected. This is the reason I rated the manuscript as partly technically sound.

Major

1. Table S6 and the CRP triage algorithm: number requiring confirmatory Xpert test was 59.4% - this is not “almost half” as reported because this is almost 60%. According to page 17, 60/226 TBpos participants with Xpert results (26.5%) would be missed. However, in Table 1, there were 183 Xpert+ and 38 Xpert- = 221 Xpert results among TBpos participants. Furthermore, it was stated that “if Xpert MTB/Rif were implemented for all without CRP triage, 38 (16.8%) participants would be missed”. This is correct if indeed the number of Xpert results is 226 but not 221 (38/221 = 17.2%). However, the sensitivity reported in Table S6 is 82.8% which means 17.2% will be missed. Please check these numbers or am I missing something? If I’m correct please also fix references to these numbers in the discussion.

Minor

2. Abstract: please use clinical or diagnostic accuracy as the term “clinical utility” is often used to refer to impact on patient outcomes.

3. Abstract: were the 765 serum samples from 765 adults presenting with respiratory symptoms or were there multiple samples from some patients? The results in the main text reported 765 participants but not clear in the abstract.

4. Abstract: is MGIT a well know acronym?

5. Abstract and elsewhere: “pooled sensitivity” The use of ‘pooled’ is confusing and misleading. The results are from a single study and not from a meta-analysis or from combining results from multiple cohorts in one study. A more appropriate word is ”overall” but only needed when there is ambiguity about whether you are referring to a subgroup or all patients.

6. Abstract and elsewhere: Please change “operator” to “operating” in “Receiver Operator Characteristic”.

7. Abstract discussion and main discussion: I can’t see the promise since it failed to meet the minimum TPP value of 90% sensitivity even in a retrospective study and results were not reported in the abstract for HIV+ individuals. Even for PLHIV, is it really promising given the poor specificity that is no better than tossing a coin when assessed against culture?

8. Study population: the study is described as a nested case control study. Is this correct since not all the cases were included unless I’m missing something?

9. Sample size calculation: “targeted performances of the index test” please specify what they are for sensitivity and specificity – were these the values in the TPP? Were the same estimates used for the HIV+ subpopulation?

10. Statistics: Suggest rephrasing the sentence “Results were reported as a point estimate with 95%CI calculated using Wilson’s method for pre-defined CRP cutoff-points” as it sort of implies Wilson’s method is for the cutoffs rather than CIs. Selecting variables based on statistical significance in univariable analysis is discouraged while the use of backward elimination is encouraged. Given the exploratory nature of the analysis and because it was pre-specified, I will not make this a major issue but please note for future analyses. How did you deal with missing data or was your analysis a complete case analysis? Was the same dataset used for the univariable and multivariable analysis as the dataset will be different depending on whether or not each variable had some missing data.

11. Results: please give the number of women so it reads “…X (40.1%) were women…”. Also give % for “…527/765…” so it reads “…527/765 (X%)…” There is a typo in “Night sweats” should be “night sweats”

12. Please give CIs for all estimates of sensitivity and specificity reported on page 14. Please write sens and spec in full on this page when referring to the TPP.

13. Figure S1_Fig did not download properly so I cannot comment on it.

14. Page 15: Please cross-reference Table S2 for the results against XRS that were reported on this page.

15. Results of subgroup analyses: results of HIV+ and HIV- patients were compared but specificity not stated for HIV positives especially as the sensitivity of >90% was highlighted. Please give the CIs along with the estimates reported in this section.

16. Inconsistent use of the terms HIV+/HIV-, HIV-positive/HIV-negative – please choose one pair and use consistently throughout. Are you using HIV+ or PLHIV?

17. Page 17: “…difference in the AUC between these two populations was not significant.” Did you formally test this? I interpret this to be the case with the use of term “not significant”.

18. Discussion: “…but fell 13% short of the specificity of the TPP…” I think you mean 13 percentage points rather than 13% because you are referring to an absolute difference rather than 13% of the TPP specificity value. Same too later in that paragraph where you mentioned “10% more”.

19. Discussion: earlier stated that the analysis by study site was pre-specified (in methods as country of origin of the sample,) but in the discussion stated as post-hoc. Which is it?

20. I couldn’t figure out why the numbers do not add up to the total for smear positive culture positive and smear negative culture positive.

21. For complete reporting, please give the number of cases in addition to n (e.g. n/N) in all main and supplementary tables like Table 3 that report sensitivities and specificities. This will enable readers to easily derive 2x2 tables if they wish to do so.

22. The STARD checklist was used to report the study but was not included as a supplementary file. Not required by the journal?

7. PLOS authors have the option to publish the peer review history of their article (what does this mean?). If published, this will include your full peer review and any attached files.

Reviewer #1: No

Reviewer #2: No

Reviewer #4: No

---

## [Author Response · Author response to Decision Letter 1]

17 May 2021

For ease of reference, the specific reviewer and editor comments and our associated replies are detailed in a table format in the document uploaded as a response to the reviewers. For ease of access, these are listed here too. 

We will address reviewers points below sequentially as raised. 

Reviewer 1:

Major points: 

Comment: of the 4 major points I raised were dismissed without substantive discussion. I have asked to have the manuscript reviewed by a statistician

Response: We apologise to the reviewer that they felt their points were dismissed without a proper discussion into issues raised. We too have had the manuscript reviewed by a statistician and our reply is as follows:

We think the methodology the reviewer proposes broadly makes sense. We agree that in large and complex analyses data-driven variable selection and correction for selection bias are important facets of sound methodology with studies such as ours. 

However, these techniques are often aimed at improving the validity of analysis concerning associations with secondary outcomes that go beyond the main outcome of disease. Furthermore, as we have previously mentioned, they are typically used to address circumstances in which cases and controls are selected with unequal probability, a situation that does not strictly apply to our study. We are also of the opinion our regression analysis is a relatively “simple” one, largely designed to be illustrative, and as such we don’t see significant additional value in overcomplicating it with the suggested methodology, especially as the number of variables included a priori was small. While our ROC curve analysis is clearly more central to the study’s results, we feel the explanation above justifies the methodology we have used.

Reviewer 2:

Minor points:

Comment: In Table 1 / Smear status, we are told there were 289 participants S+C+ and 102 S-C+, whereas in Table 3 to the left of S+C+ we are given n=663 and for S-C+ we are given n=476; these numbers are not the numbers for S+C+ and S-C+ and must refer to [S+C+ plus all TB test neg] and [S-C+ plus all TB neg] but this is not immediately clear on first reading (or second reading I confess) - it might help to explicitly state in the line just below the table "Smear-positive and smear-negative participants were each assessed against all TBneg (n= 374)", or to insert another line in the table for All TB test neg

Response: We agree with the reviewer that this is not clear and have amended the legend below the table to include how these numbers were arrived at, in line with the reviewer’s recommendations. 

Comment: In the Discussion you note (in lines 306-309) that using a CRP cut off of 8 mg/L would result in approximately 26% of culture positive cases being missed. I expect this 26% to accord with the numbers given in Supplementary Figure 1 which relates to the use of the 8 mg/L cut off. In S Fig 1 it appears that a total of 79 culture + cases occurred among those with CRP <8 mg/L (with the total of all culture positive cases being 79 + 312 = 391) which looks like 20.2% (79/391) were missed (because CRP <8). Where does 26% come from? Is there a typographical error and you mean to continue referring to an 18 mg/L cut off, not an 8 mg/L cut off as the text currently states? (In that case it would obviously have nothing to do with S Fig 1.)

Reponse: We agree with the reviewer that these numbers do not concord with each other. 

The reason for the use of 26% was because this statement in the discussion related to the use of the Xpert reference standard not the microbiological reference standard – 26% was calculated on the basis of the number of participants that had been tested with Xpert MTB/Rif who would have been missed diagnoses according to the use of the algorithm specified in the “CRP Triage Algorithm” section of the results. We have amended the discussion to make this clearer. 

However, the value of 26% itself was incorrect – as pointed out by a separate reviewer, we mistakenly included indeterminate Xpert results in the denominator to arrive at this value. For all other analysis in this paper, we had excluded indeterminate Xpert results (our intention). We have clarified this intention in the methods and amended all relevant areas in the results and discussion to correct this error. We apologise for the confusion and hope these amendments address the reviewer’s comment. 

Reviewer 4

Major points:

Comment: Table S6 + CRP Triage Algorithm - The number requiring confirmatory Xpert test was 59.4% - this is not “almost half” as reported because this is almost 60%. According to page 17, 60/226 TBpos participants with Xpert results (26.5%) would be missed. However, in Table 1, there were 183 Xpert+ and 38 Xpert- = 221 Xpert results among TBpos participants. Furthermore, it was stated that “if Xpert MTB/Rif were implemented for all without CRP triage, 38 (16.8%) participants would be missed”. This is correct if indeed the number of Xpert results is 226 but not 221 (38/221 = 17.2%). However, the sensitivity reported in Table S6 is 82.8% which means 17.2% will be missed. Please check these numbers or am I missing something? If I’m correct please also fix references to these numbers in the discussion

Response: Regarding the reviewer’s first point, the authors intention was to convey that almost half of participants did not require a confirmatory test. However, given that this may be deemed inaccurate, we have changed the manuscript to read that 40% (100-59.4% = 40.6% ) did not require confirmatory testing. We hope this addresses the reviewer’s first point. 

Regarding the second half of the reviewer’s comment, the confusion arises due to indeterminate Xpert MTB/Rif test results. In 5 cases where the Xpert result was indeterminate, the culture result was positive, hence the differing denominators of 221 in the table but 226 on page 17. The use of 226 as the denominator was a mistake as we decided to exclude indeterminate Xpert results from the analyses we completed on Xpert performance. As a result, we have changed the relevant parts of the manuscript to reflect the numbers and percentages that are incorrect. We have also added a sentence to the Methods section to clarify how indeterminate Xpert MTB/Rif results were handled.

Minor points:

Comment: Abstract - please use clinical or diagnostic accuracy as the term “clinical utility” is often used to refer to impact on patient outcomes

Reponse: The authors agree with the reviewer and have made the suggested change

Comment: Abstract - were the 765 serum samples from 765 adults presenting with respiratory symptoms or were there multiple samples from some patients? The results in the main text reported 765 participants but not clear in the abstract

Response: Each sample was from a single study participant. The authors have amended the wording in the abstract to reflect this.

Comment: Abstract - Is MGIT a well know acronym?

Response: MGIT is the most commonly used mycobacterial liquid culture medium, but the authors recognise that best practice is define all acronyms at first usage and so have modified the abstract to replace “MGIT” with “liquid” in line with the reviewer’s comment. 

Comment: Abstract + elsewhere - “pooled sensitivity” The use of ‘pooled’ is confusing and misleading. The results are from a single study and not from a meta-analysis or from combining results from multiple cohorts in one study. A more appropriate word is ”overall” but only needed when there is ambiguity about whether you are referring to a subgroup or all patients 

Response: We agree with the reviewer and have changed all relevant sections of the manuscript

Comment: Abstract - Please change “operator” to “operating” in “Receiver Operator Characteristic”

Response: We agree with the reviewer and have changed all relevant sections of the manuscript

Comment: Abstract discussion and main discussion - I can’t see the promise since it failed to meet the minimum TPP value of 90% sensitivity even in a retrospective study and results were not reported in the abstract for HIV+ individuals. Even for PLHIV, is it really promising given the poor specificity that is no better than tossing a coin when assessed against culture?

Response: We agree with the reviewer that overall performance of the test was suboptimal with respect to the MRS and the target TPP. However, when different CRP cutoff-points were explored, performance of the test in HIV positive participants did near TPP; at a cutoff-point of 23mg/L sensitivity was 85.6% and specificity was 69.6% for HIV+ participants as we stated in the subgroup section of the results. Further to this, were an Xpert reference standard to be used as would be more likely in many low-income settings, performance of CRP relative the TPP improves albeit due to a reduction in sensitivity of the confirmation test. Our supplementary data (Table S3) shows that at a cutoff-point of 18mg/L using the XRS, overall sensitivity and specificity were 84.7% and 69.5% respectively. Using difference cutoff-points for HIV+ and HIV- participants produced similar performances in both subgroups. We would argue that these results are sufficiently close enough to the TPP to warrant further investigation into the role of CRP as a triage test in this cohort of patients and to justify our assertion that they show promise. 

Comment: Study population - the study is described as a nested case control study. Is this correct since not all the cases were included unless I’m missing something?

Response: The authors feel that the present study does qualify as a nested case-control design. While it is very true that nested case-control designs often include all cases from the parent cohort as the reviewer states, this is not necessary to define a case-control design. The key defining characteristic of a nested case-control study is that a random sample of cases and controls are selected from a fully enumerated cohort, which is the case here. This is supported in Rothman KJ, Greenland S, Lash TL, editors. Modern epidemiology. Lippincott Williams & Wilkins; 2008.

Comment: Sample size calculation - “targeted performances of the index test” please specify what they are for sensitivity and specificity – were these the values in the TPP? Were the same estimates used for the HIV+ subpopulation?

Response: We agree and have made the changes suggested in the reviewer’s comment

Comment: Statistics - Suggest rephrasing the sentence “Results were reported as a point estimate with 95%CI calculated using Wilson’s method for pre-defined CRP cutoff-points” as it sort of implies Wilson’s method is for the cutoffs rather than CIs. Selecting variables based on statistical significance in univariable analysis is discouraged while the use of backward elimination is encouraged. Given the exploratory nature of the analysis and because it was pre-specified, I will not make this a major issue but please note for future analyses. How did you deal with missing data or was your analysis a complete case analysis? Was the same dataset used for the univariable and multivariable analysis as the dataset will be different depending on whether or not each variable had some missing data.

Response: We agree with the reviewer on the first point regarding the structure of the quoted sentence and have amended the manuscript to make this clearer. 

We note the reviewer’s comments surrounding the univariable analysis with thanks and we will incorporate this advice into future analyses. 

Our analysis was a complete case analysis, with the same dataset used for both univariable and multivariable analyses. We have amended the methods section to clarify this.

Comment: Result - please give the number of women so it reads “…X (40.1%) were women…”. Also give % for “…527/765…” so it reads “…527/765 (X%)…” There is a typo in “Night sweats” should be “night sweats”

Response: We agree with the reviewer and have amended these points

Comment: Page 14 - Please give CIs for all estimates of sensitivity and specificity reported on page 14. Please write sens and spec in full on this page when referring to the TPP.

Response: We agree with the reviewer and have amended the manuscript with the suggested changes. 

Comment: Figure S1_Fig did not download properly so I cannot comment on it

Response: We will attempt to re-upload this file to see if it assists the reviewer in downloading it. 

Comment: Page 15 - Please cross-reference Table S2 for the results against XRS that were reported on this page

Response: We agree with the reviewer and have amended this

Comment: Results – subgroup analysis - results of HIV+ and HIV- patients were compared but specificity not stated for HIV positives especially as the sensitivity of >90% was highlighted. Please give the CIs along with the estimates reported in this section.

Response: We agree with the reviewer and have amended this section of the results to include 95%CI and the requested specificities from Table 3.

Comment: Throughout - Inconsistent use of the terms HIV+/HIV-, HIV-positive/HIV-negative – please choose one pair and use consistently throughout. Are you using HIV+ or PLHIV?

Response: We agree with the reviewer that our use of HIV+ vs positive etc is inconsistent and so we have amended the manuscript to use the nomenclature HIV+ and HIV-. 

Comment: Page 17.- “…difference in the AUC between these two populations was not significant.” Did you formally test this? I interpret this to be the case with the use of term “not significant”.

Response: We did formally test this but did not report the p value for brevity’s sake. 

Comment: Discussion - “…but fell 13% short of the specificity of the TPP…” I think you mean 13 percentage points rather than 13% because you are referring to an absolute difference rather than 13% of the TPP specificity value. Same too later in that paragraph where you mentioned “10% more”

Response: We did mean percentage points and not percent and thank the reviewer for pointing this out. We have changed the relevant parts of the manuscript. 

Comment: Discussion - earlier stated that the analysis by study site was pre-specified (in methods as country of origin of the sample,) but in the discussion stated as post-hoc. Which is it?

Response: Analysis by country of origin was post-hoc and the methods section was incorrect. We have amended the methods to reflect this. 

Comment: Results/Table - I couldn’t figure out why the numbers do not add up to the total for smear positive culture positive and smear negative culture positive.

Response: We have amended the legend to the relevant table to make it clearer as to how the figures were reached. 

Comment: Table 3 and supplementary material - For complete reporting, please give the number of cases in addition to n (e.g. n/N) in all main and supplementary tables like Table 3 that report sensitivities and specificities. This will enable readers to easily derive 2x2 tables if they wish to do so.

Response: We agree and have amended Table 3 and the supplementary tables S2 and S4 in line with the reviewer’s suggestion. 

Comment: Supplementary material - The STARD checklist was used to report the study but was not included as a supplementary file. Not required by the journal?

Response: The reviewer is correct that this was not required by the journal. However, we are happy to upload the guidelines as a supplementary file.

---

## [Decision Letter · Decision Letter 2]

2 Jun 2021

PONE-D-20-19334R2

Evaluation of the diagnostic performance of laboratory-based c-reactive protein as a triage test for active pulmonary tuberculosis

PLOS ONE

Dear Dr. Schumacher,

Thank you for submitting your manuscript to PLOS ONE. After careful consideration, we feel that it has merit but does not fully meet PLOS ONE’s publication criteria as it currently stands. Therefore, we invite you to submit a revised version of the manuscript that addresses the points raised during the review process.

Please submit your revised manuscript. If you will need significantly more time to complete your revisions, please reply to this message or contact the journal office at plosone@plos.org. Please include the following items when submitting your revised manuscript:

We look forward to receiving your revised manuscript.

Kind regards,

Frederick Quinn

Academic Editor

PLOS ONE

Journal Requirements:

Reviewers' comments:

Reviewer's Responses to Questions

**Comments to the Author**

1. If the authors have adequately addressed your comments raised in a previous round of review and you feel that this manuscript is now acceptable for publication, you may indicate that here to bypass the “Comments to the Author” section, enter your conflict of interest statement in the “Confidential to Editor” section, and submit your "Accept" recommendation.

Reviewer #1: All comments have been addressed

Reviewer #2: All comments have been addressed

Reviewer #4: (No Response)

2. Is the manuscript technically sound, and do the data support the conclusions?

Reviewer #1: Yes

Reviewer #2: Yes

Reviewer #4: Yes

3. Has the statistical analysis been performed appropriately and rigorously? 

Reviewer #1: Yes

Reviewer #2: Yes

Reviewer #4: Yes

4. Have the authors made all data underlying the findings in their manuscript fully available?

Reviewer #1: Yes

Reviewer #2: Yes

Reviewer #4: Yes

5. Is the manuscript presented in an intelligible fashion and written in standard English?

Reviewer #1: Yes

Reviewer #2: Yes

Reviewer #4: Yes

6. Review Comments to the Author

Reviewer #1: (No Response)

Reviewer #2: The clarity of the paper is much improved in this version. I would recommend acceptance but have chosen 'minor revision' only because there are 3 small errors noted:

(1) Line 28: Since the word data is the plural of datum, the sentence should read "...limited data are available..."

(2) Line 317: the word 'quarter' is misspelled (appears as quater).

(3) Line 372-3: Since 'data' is plural, the sentence should read "Previous data assessing the role of CRP in the diagnosis of TB have largely been...", not 'has largely been'.

Reviewer #4: Thank you for addressing all of my comments appropriately. I have a few typos I spotted which are due to the edits you made.

1. Where you have replaced "a pooled" with "a overall", please correct to "an overall".

2. Discussion: in the results you gave 40.6% but stated 40% in the discussion. Please write "about 40%" of give exact value of 40.6%.

3. Discussion: "quater" should be "quarter".

7. PLOS authors have the option to publish the peer review history of their article (what does this mean?). If published, this will include your full peer review and any attached files.

Reviewer #1: No

Reviewer #2: No

Reviewer #4: No

---

## [Author Response · Author response to Decision Letter 2]

7 Jun 2021

For ease of reference, the specific reviewer and editor comments and our associated replies are detailed in a table format in the document uploaded as a response to the reviewers. For ease of access, these are listed here too. 

We will address reviewers points below sequentially as raised. 

Reviewer 2:

Comment: Line 28: Since the word data is the plural of datum, the sentence should read "...limited data are available..."

Response: We agree with the reviewer and have corrected this grammar in the abstract

Comment: Line 317: the word 'quarter' is misspelled (appears as quater).

Response: We thank the reviewer for bring our attention to this! We have changed the spelling.

Comment: Line 372-3: Since 'data' is plural, the sentence should read "Previous data assessing the role of CRP in the diagnosis of TB have largely been...", not 'has largely been'.

Response: We agree with the reviewer and have corrected this grammar.

Reviewer 4:

Comment: Where you have replaced "a pooled" with "a overall", please correct to "an overall"

Response: Many thanks to the reviewer for bringing this to our attention. We have made this correction to the relevant points in the manuscript have corrected a similar grammatical mistake we identified elsewhere. 

Comment: Discussion: in the results you gave 40.6% but stated 40% in the discussion. Please write "about 40%" of give exact value of 40.6%.

Response: We agree with the reviewer and have altered the relevant bit of the discussion to read “about 40%”.

Comment: Discussion: "quater" should be "quarter".

Response: Similar to point 1, thanks again to the reviewer for bringing this to our attention. Overall, a lesson in making sure one does a final spell check!

---

## [Decision Letter · Decision Letter 3]

18 Jun 2021

Evaluation of the diagnostic performance of laboratory-based c-reactive protein as a triage test for active pulmonary tuberculosis

PONE-D-20-19334R3

Dear Dr. Schumacher,

We’re pleased to inform you that your manuscript has been judged scientifically suitable for publication and will be formally accepted for publication once it meets all outstanding technical requirements.

Kind regards,

Frederick Quinn

Academic Editor

PLOS ONE

Additional Editor Comments (optional):

Reviewers' comments:

Reviewer's Responses to Questions

**Comments to the Author**

1. If the authors have adequately addressed your comments raised in a previous round of review and you feel that this manuscript is now acceptable for publication, you may indicate that here to bypass the “Comments to the Author” section, enter your conflict of interest statement in the “Confidential to Editor” section, and submit your "Accept" recommendation.

Reviewer #2: All comments have been addressed

Reviewer #4: All comments have been addressed

2. Is the manuscript technically sound, and do the data support the conclusions?

Reviewer #2: Yes

Reviewer #4: (No Response)

3. Has the statistical analysis been performed appropriately and rigorously? 

Reviewer #2: Yes

Reviewer #4: (No Response)

4. Have the authors made all data underlying the findings in their manuscript fully available?

Reviewer #2: Yes

Reviewer #4: (No Response)

5. Is the manuscript presented in an intelligible fashion and written in standard English?

Reviewer #2: Yes

Reviewer #4: (No Response)

6. Review Comments to the Author

Reviewer #2: All of the outstanding corrections have been made. In my opinion, the manuscript should be accepted now.

Reviewer #4: (No Response)

7. PLOS authors have the option to publish the peer review history of their article (what does this mean?). If published, this will include your full peer review and any attached files.

Reviewer #2: No

Reviewer #4: No

---

## [Editor Report · Acceptance letter]

2 Jul 2021

PONE-D-20-19334R3 

Evaluation of the diagnostic performance of laboratory-based c-reactive protein as a triage test for active pulmonary tuberculosis 

Dear Dr. Schumacher:

I'm pleased to inform you that your manuscript has been deemed suitable for publication in PLOS ONE. Congratulations! Your manuscript is now with our production department. 

Kind regards, 

on behalf of

Dr. Frederick Quinn 

Academic Editor

PLOS ONE